# LADA: Enabling Adaptation of Black-Box LLMs to Dynamic Domain Changes at Test Time

## Abstract

Test-time adaptation (TTA) for black-box large language models (LLMs) seeks to adapt models to target-domain inputs during testing to address distribution shifts, without accessing model parameters. Most existing approaches rely on adapters trained with substantial target-domain data—often scarce or unreliable—and these adapters are tightly coupled to their training distribution, degrading in dynamic scenarios with changing domains. To solve this, we propose LADA (Learning-to-ADapt for black-box LLMs), a framework leveraging meta-training for continuous, rapid adaptation to unseen, dynamic domains. Specifically, LADA meta-trains an adapter on diverse tasks from multiple datasets (covering semantic clusters and error types) to learn transferable adaptation skills. At test time, the adapter needs only a few target-domain examples for lightweight adaptation and guides the LLM stepwise toward domain-appropriate reasoning via adaptive selection of reliable steps—no LLM parameter updates required. Experiments on various benchmark datasets validate the effectiveness of the proposed approach.

## 1 Introduction

Large language models (LLMs), such as OpenAI's GPT series (OpenAI, 2023) and Meta's Llama family (Touvron et al., 2023), have demonstrated remarkable capabilities in language understanding, reasoning, and generation through training on vast text corpora. These models are now being deployed across diverse real-world applications—including software engineering (Fan et al., 2023; Hou et al., 2024), healthcare (Wang & Zhang, 2024; Bedi et al., 2025), and legal assistance (Huang et al., 2023a; Zhou et al., 2024)—where they often encounter inputs that deviate significantly from their pretraining distribution.

To address this distribution shift, test-time adaptation (TTA) has been adopted for improving model robustness by adapting to target-domain inputs during inference (Hu et al., 2025a). A growing body of work focuses on white-box TTA methods, which update internal model parameters using unlabeled test data via fine-tuning (Hardt & Sun, 2024; Hübotter et al., 2025) or perplexity minimization (Hu et al., 2025a). While effective, these approaches require full access to model weights and incur high computational costs, making them incompatible with black-box, API-served LLMs widely used in practice.

In response, recent studies explore black-box TTA strategies that leverage auxiliary adapters or scorers to guide the LLM without modifying its parameters. For example, (Liu et al., 2024) use a small trainable model to adjust next-token probabilities, while (Shi et al., 2024) and (Sun et al., 2024b) employ rerankers or scoring models to select among multiple generated candidates. However, these methods share a critical limitation: they rely on substantial target-domain data for adapter training, resulting in domain-tied adapters that fail in dynamic real-world settings—where task requirements and data distributions shift (e.g., changing customer service intents, evolving medical guidelines, emerging coding domains). Re-collecting domain-specific data and retraining adapters for every new context is impractical, and assuming prior knowledge of future domains contradicts the open-ended nature of dynamic deployment environments.

To overcome this challenge, we propose LADA (Learning-to-ADapt for black-box LLMs), a novel framework enabling continuous, rapid adaptation to unseen, changing domains at test time—even

without prior domain information. Unlike approaches with static, domain-specific adaptations, LADA meta-trains an adapter across diverse tasks (spanning datasets, semantic clusters, error types) to learn transferable skills, letting it quickly adapt to new domains via few in-context examples at inference. At test time, the adapter uses a small set of current-domain positive–negative reasoning pairs for lightweight adaptation, then guides the frozen LLM step-by-step to select reliable reasoning steps, steering generation toward domain-appropriate trajectories without parameter updates or prior domain exposure. Critically, its cross-domain and error-mode generalization ensures effectiveness as the target domain evolves.

- We propose LADA, a meta-learning-based framework for black-box LLMs that enables continuous test-time adaptation to dynamic, changing domains (rather than a single fixed domain), eliminating reliance on prior knowledge of specific target domains.
- LADA avoids the need for substantial training data from a specific target domain; instead, it meta-trains an adapter on diverse tasks from multiple datasets and only requires a few target-domain examples for lightweight adaptation at test time.
- We theoretically prove that our adaptive selection TTA policy for black-box LLMs would obtain equal or higher cumulative reward expectation than the baseline policy that naively accepts the first sampled reasoning step at test time with mild assumptions.

## 2 RELATED WORK

**Traditional TTA.** Deep neural networks often experience performance degradation when there is a distribution shift between training and test data (Quiñonero-Candela et al., 2022). Test-time adaptation (TTA) (Wang et al., 2024; Liang et al., 2025) has emerged as a promising paradigm to mitigate this problem by adapting a pre-trained model to unlabeled test data prior to prediction. Some TTA methods achieve adaptation through entropy minimization. For example, Wang et al. (2021) optimize batch normalization layers by minimizing the entropy of predictions, while Niu et al. (2022) build on this by focusing on reliable and non-redundant samples, and Zhang et al. (2025) further extend it with a conservative strategy for unreliable samples. Other approaches perform self-training with pseudo-labels: Goyal et al. (2022) utilize a specialized soft label termed the conjugate pseudo-label, Sun et al. (2024a) construct pseudo-labels via label propagation, and Hu et al. (2025b) employ candidate pseudo-labels to refine the model. A further line of work adapts models using consistency information, for example, by enforcing consistency among neighboring samples (Jang et al., 2023), with class prototypes (Wang et al., 2023a), or between the current model and a teacher model (Döbler et al., 2023). While these approaches have shown effectiveness in classification tasks, they do not transfer well to LLMs and often fail when applied directly (Hu et al., 2025a).

**LLM TTA.** In general, this paradigm can be categorized into white-box and black-box settings, depending on whether LLM parameters are accessible. In the white-box setting, the model parameters are updated using the target examples. For each test input, Hardt & Sun (2024) retrieve its neighbors and fine-tune the model on their text at test time. Building on this idea, Hübotter et al. (2025) further reduce redundancy among the selected data to optimize the overall information gain of the chosen examples. Along similar lines, (Yu et al., 2025) enable LLM to retrieve and learn from related reasoning traces to enhance reasoning capabilities. More recently, Hu et al. (2025a) reveal that more accurate predictions can be obtained by minimizing the input perplexity of unlabeled test data. Yet, these approaches incur substantial computational and memory overhead and are inapplicable to today's API-served LLMs, where parameters are inaccessible, thereby limiting their practicality in real-world scenarios.

For black-box TTA, model parameters are inaccessible. Some methods assume access to the output token probabilities of the LLM and perform adaptation by correcting the probability distribution of the next token. For example, Huang et al. (2023b) adaptively interpolate the LLM's prediction probabilities with retrieval results from a datastore. Ormazabal et al. (2023) and Liu et al. (2024) leverage a smaller model fine-tuned on target-domain data to correct the LLM's next-token probabilities using its context-dependent predictions during inference. It is worth noting that certain LLMs, such as Claude models (Anthropic, 2025), do not expose token-level probabilities, thereby restricting the applicability of such methods. Another line of work relies only on observable outputs generated by LLM. They typically leverage target-domain training data to train a scoring model that ranks candidate responses through multiple sampling. For instance, Khalifa et al. (2023) use

a step-level discriminator to select the best reasoning step, Sun et al. (2024b) employ an adapter to guide sentence-level beam search, and Shi et al. (2024) utilize a reranker to rank complete solutions, thereby enabling adaptation to the target domain. However, these approaches assume access to substantial target-domain data, which is not always feasible in practical scenarios.

# 3 PROPOSED METHOD

We begin by introducing the notations used throughout the paper and outlining the problem setting. In step-by-step reasoning tasks such as question answering or problem solving, given an input question $q$, a black-box LLM $\pi_{\text{LLM}}$ incrementally generates a sequence of $l$ intermediate reasoning steps $\boldsymbol{r} = (r_1, \ldots, r_l)$. Let $\text{Sample}\,(p, t)$ denote the $t$-th trial to sample the next sentence from distribution $p$. The generation of each reasoning step usually follows the standard sampling process:

$$r_s = \text{Sample}\,(\pi_{\text{LLM}}(\cdot \mid q, r_1, \ldots, r_{s-1}), 1)\,. \tag{1}$$

When $\pi_{\text{LLM}}$ is applied to a new target domain, the distribution of input questions $p_T(q)$ often differs from the distribution $p_S(q)$ seen during pretraining. This distribution shift typically leads to degraded reasoning quality and reduced accuracy. To address this problem, we introduce a meta-trained adapter $f_{\boldsymbol{\theta}}$ that acts as a step-level scoring model, and each step is generated as:

$$r_s = \sum_{t=1}^{t^*} \mathbb{1}[f_{\boldsymbol{\theta}}(\text{Sample}(\pi_{\text{LLM}}(\cdot \mid q, r_1, \ldots, r_{s-1}), t)) > \tau] \\ \cdot \text{Sample}(\pi_{\text{LLM}}(\cdot \mid q, r_1, \ldots, r_{s-1}), t)\,, \tag{2}$$

where $t^*$ denotes the first sample index $t$ such that $f_{\boldsymbol{\theta}}\,(\text{Sample}\,(\pi_{\text{LLM}}\,(q, r_1, \ldots, r_{s-1}), t)) > \tau$, with $\tau$ a predefined threshold. To ensure efficiency, we set a maximum number of trials $t_{\max}$. If no sample satisfies this condition within $t_{\max}$ trials, the step with the highest score is selected. By iteratively applying this rule, the adapter guides $\pi_{\text{LLM}}$ step by step, selecting reliable reasoning steps and avoiding potential errors induced by distribution shift, thereby steering its reasoning toward domain-appropriate trajectories.

## 3.1 OVERVIEW

LADA achieves test-time adaptation for black-box LLMs using an adapter that is meta-trained in advance. To meta-train the adapter, we construct tasks via the Cartesian product of partitioned semantic clusters and specified error types, with each task containing step-level positive–negative reasoning pairs with the same semantic topic and error type. Then these tasks are utilized to meta-train the adapter as a step-level scorer in an inner–outer loop: the inner loop adapts to each task with a few samples, while the outer loop optimizes meta-parameters to capture transferable patterns across diverse tasks, thereby enabling effective adaptation to dynamic domain changes. At test-time, the adapter is rapidly optimized with a few paired examples from the target domain and guides the black-box LLM stepwise by adaptively retaining reliable reasoning steps while resampling unreliable ones, thereby steering the generation process toward domain-appropriate reasoning trajectories.

## 3.2 THE LADA FRAMEWORK

### 3.2.1 TASK CONSTRUCTION

Our goal is to construct diverse meta-training tasks that simulate potential adaptation cases and enable the adapter to capture transferable patterns across tasks. Owing to their accessibility and cross-domain coverage, we leverage multiple publicly available datasets $\{\mathcal{D}_i\}_{i=1}^a$, where each dataset is defined as $\mathcal{D}_i = \{(q_j, \boldsymbol{r}_j)\}_{j=1}^{b_i}$, with $a$ denoting the number of datasets and $b_i$ the number of examples in the $i$-th dataset.

To ensure semantic diversity across tasks, we first cluster examples according to their semantic similarity. Specifically, we obtain embeddings of all $q$ in each $\mathcal{D}_i$ using an embedding model and apply $k$-means to partition $\mathcal{D}_i$ into $k_i$ clusters. In total, we obtain $m = \sum_{i=1}^a k_i$ clusters from all datasets, denoted as $\{\mathcal{C}_i\}_{i=1}^m$. Each cluster is of the form $\mathcal{C}_i = \{(q_j, \boldsymbol{r}_j)\}_{j=1}^{n_i}$, where $\boldsymbol{r}_j = (r_{j,s})_{s=1}^{l_j}$.

When distribution shifts occur in input questions, $\pi_{\text{LLM}}$ may produce various reasoning errors, such as hallucination, repetition, or missing steps. To capture this variety and ensure diversity of error types across tasks, we formalize them into $h$ categories, denoted $\{e_g\}_{g=1}^h$. In order for the adapter to recognize these errors, we employ an oracle LLM $\pi_{\text{oracle}}$ to synthesize positive–negative reasoning pairs for each error type $e_g$ in three steps. Given an example $(q_j, \boldsymbol{r}_j)$, $\pi_{\text{oracle}}$ first selects a step $r_{j,s_g}$ from $\boldsymbol{r}_j$ that is susceptible to the error type $e_g$. Conditioned on $(q_j, \boldsymbol{r}_j)$ and the chosen step $r_{j,s_g}$, $\pi_{\text{oracle}}$ then produces two corresponding variants: a correct argumentation $r_{j,s_g}^o$ that augment the step, and an erroneous counterpart $r_{j,s_g}^{e_g}$ that distorts it. This process can be formalized as:

$$\begin{cases} r_{j,s_g}^o \sim \pi_{\text{oracle}}\left(\cdot \,\middle|\, (q_j, \boldsymbol{r}_j), r_{j,s_g}, \langle o \rangle\right) \\ r_{j,s_g}^{e_g} \sim \pi_{\text{oracle}}\left(\cdot \,\middle|\, (q_j, \boldsymbol{r}_j), r_{j,s_g}, \langle e_g \rangle\right) \end{cases}, \tag{3}$$

where $\langle o \rangle$ and $\langle e_g \rangle$ denote the prompts for correct argumentation and error generation of type $e_g$, respectively. Then by truncating the reasoning sequence after $r_{j,s_g}$, we construct a positive–negative reasoning pair $(\boldsymbol{x}_j^+, \boldsymbol{x}_j^-)$, where $\boldsymbol{x}_j^+ = (q_j, r_{j,1}, \ldots, r_{j,s_g-1}, r_{j,s_g}^o)$ and $\boldsymbol{x}_j^- = (q_j, r_{j,1}, \ldots, r_{j,s_g-1}, r_{j,s_g}^{e_g})$. The two reasoning sequences differ only in their final step, and continuing reasoning from $\boldsymbol{x}_j^-$ typically leads $\pi_{\text{LLM}}$ to fail in producing the correct answer. This design isolates the impact of a single step correctness, allowing the adapter to focus on step-level scoring while avoiding interference from later steps.

Finally, taking the Cartesian product of the $m$ clusters and the $h$ error types yields $mh$ tasks, where each task is defined by the same semantic topic and consistent error type in the final reasoning step. Formally, the meta-learning task set is $\mathcal{M} = \{\mathcal{T}_i\}_{i=1}^u$ with $u = mh$, where each task $\mathcal{T}_i = \{(\boldsymbol{x}_j^+, \boldsymbol{x}_j^-)\}_{j=1}^{v_i}$ consists of $v_i$ positive–negative pairs.

### 3.2.2 META-TRAINING

Building on the constructed task set, we employ a meta-training (Finn et al., 2017) stage to enable $f_{\boldsymbol{\theta}}$ to be quickly adapted when only a few target-domain examples are available. In each meta-training iteration, we first sample $z$ tasks $\{\mathcal{T}_i\}_{i=1}^z$ from the task set $\mathcal{M}$. For a given task $\mathcal{T}_i$, a support set $\mathcal{S}_i = \{(\boldsymbol{x}_j^+, \boldsymbol{x}_j^-)\}_{j=1}^{c_i}$ of $c_i$ positive–negative pairs is sampled for inner-loop adaptation, while a query set $\mathcal{Q}_i = \{(\boldsymbol{x}_j^+, \boldsymbol{x}_j^-)\}_{j=1}^{d_i}$ of $d_i$ pairs is held out to evaluate generalization in the outer loop, with $\mathcal{S}_i \cap \mathcal{Q}_i = \varnothing$.

In the inner loop, the goal is to obtain the task-specific adapter for $\mathcal{T}_i$. To this end, the base adapter $f_{\boldsymbol{\theta}}$ is adapted by minimizing the inner loss $\mathcal{L}^{\text{in}}$ on the corresponding support set $\mathcal{S}_i$:

$$\min_{\boldsymbol{\theta}} \; \mathcal{L}^{\text{in}}(f_{\boldsymbol{\theta}}, \mathcal{S}_i), \tag{4}$$

which simulates the adaptation to an unseen target task with a few examples. The resulting task-specific parameters are obtained through the update:

$$\boldsymbol{\theta}_i' \leftarrow \boldsymbol{\theta} - \alpha \nabla_{\boldsymbol{\theta}} \mathcal{L}^{\text{in}}(f_{\boldsymbol{\theta}}, \mathcal{S}_i), \tag{5}$$

where $\alpha$ is the adaptation step size.

Subsequently, the outer loop evaluates the generalization of the adapted parameters $\boldsymbol{\theta}_i'$ on the corresponding query set $\mathcal{Q}_i$ through the outer loss $\mathcal{L}^{\text{out}}$. By minimizing the aggregated $\mathcal{L}^{\text{out}}$ across all query sets $\{\mathcal{Q}_i\}_{i=1}^z$, we obtain the meta-objective:

$$\min_{\boldsymbol{\theta}} \sum_{i=1}^z \mathcal{L}^{\text{out}}\left(f_{\boldsymbol{\theta}_i'}, \mathcal{Q}_i\right). \tag{6}$$

Since each $\boldsymbol{\theta}_i'$ is derived from $\boldsymbol{\theta}$ through the inner update, the meta-objective's dependence on $\boldsymbol{\theta}_i'$ implicitly links the optimization to $\boldsymbol{\theta}$. Therefore, the meta-parameters are updated as:

$$\boldsymbol{\theta} \leftarrow \boldsymbol{\theta} - \beta \, \nabla_{\boldsymbol{\theta}} \sum_{i=1}^z \mathcal{L}^{\text{out}}(f_{\boldsymbol{\theta}_i'}, \mathcal{Q}_i), \tag{7}$$

where $\beta$ is the meta step size. Specifically, the meta-update relies on a meta-gradient calculated over the performance on a variety of query sets. This meta-gradient averages information from

multiple tasks, thereby preventing the model from overfitting to any single task's data. Consequently, the meta-parameters are steered towards a general-purpose initialization that captures the shared structure across different tasks, which is key for acquiring transferable knowledge.

In practice, the adapter produces a score in the range $(0, 1)$, interpreted as the probability that a reasoning step is correct. The inner objective is defined using the binary cross-entropy loss:

$$\mathcal{L}^{\text{in}}(f_{\boldsymbol{\theta}}, \mathcal{S}_i) = -\frac{1}{c_i} \sum_{j=1}^{c_i} \left[ \log f_{\boldsymbol{\theta}}\left(\boldsymbol{x}_j^+\right) + \log\left(1 - f_{\boldsymbol{\theta}}\left(\boldsymbol{x}_j^-\right)\right) \right]. \tag{8}$$

For the outer objective, we adopt a max-margin loss:

$$\mathcal{L}^{\text{out}}\left(f_{\boldsymbol{\theta}_i'}, \mathcal{Q}_i\right) = \frac{1}{d_i} \sum_{j=1}^{d_i} \max\left(0, \zeta - \left[f_{\boldsymbol{\theta}_i'}\left(x_j^+\right) - f_{\boldsymbol{\theta}_i'}\left(x_j^-\right)\right]\right), \tag{9}$$

where $\zeta$ denotes the margin hyperparameter. Binary cross-entropy loss in the inner loop provides absolute probabilistic supervision, serving as a stable anchor for task-specific adaptation, whereas the margin loss in the outer loop promotes relative ranking, encouraging larger margins between correct and incorrect pairs so that the learned parameters transfer more robustly to new tasks.

Through meta-learning across diverse tasks, the adapter acquires an initialization $\boldsymbol{\theta}$ that encodes cross-task patterns, allowing it to adapt efficiently to unseen domains with limited data and remain effective in dynamic real-world scenarios.

### 3.2.3 TEST-TIME ADAPTION

Building on the meta-trained adapter, we proceed to describe the TTA process, consisting of lightweight adaptation and stepwise adaptive selection. To generate responses for a question $q \sim p_T(q)$, we first exploit a small set of paired examples $\mathcal{B}_T = \left\{(\boldsymbol{x}_j^+, \boldsymbol{x}_j^-)\right\}_{j=1}^{w}$ from the target domain, to quickly adapt the meta-trained adapter $f_{\boldsymbol{\theta}}$ by minimizing the inner loss:

$$\hat{\boldsymbol{\theta}} \leftarrow \boldsymbol{\theta} - \gamma \nabla_{\boldsymbol{\theta}} \mathcal{L}^{\text{in}}(f_{\boldsymbol{\theta}}, \mathcal{B}_T), \tag{10}$$

where $\gamma$ is the adaptation step size. Through few-shot adaptation, $f_{\hat{\boldsymbol{\theta}}}$ acquires the ability to score reasoning steps in accordance with the target domain, then it steers $\pi_{\text{LLM}}$ stepwise toward domain-appropriate reasoning trajectories.

At reasoning step $s$ for question $q$, $\pi_{\text{LLM}}$ samples one candidate step $r_s^t$ at a time, conditioned on the question $q$ and all previously accepted steps $(\hat{r}_1, \cdots, \hat{r}_{s-1})$:

$$r_s^t = \text{Sample}\left(\pi_{\text{LLM}}(\cdot \mid q, \hat{r}_1, \ldots, \hat{r}_{s-1}), t\right), \tag{11}$$

where $t$ is the index of the trial. Then the sampled candidate step $r_s^t$ is evaluated by the adapter $f_{\hat{\boldsymbol{\theta}}}$:

$$y_s^t = f_{\hat{\boldsymbol{\theta}}}\left(q_j, \hat{r}_1, \ldots, \hat{r}_{s-1}, r_s^t\right), \tag{12}$$

where $y_s^t$ denote the score for $r_s^t$. Based on these scores, we devise an adaptive selection strategy.

During the sampling–scoring process, if a candidate step $r_s^t$ obtains a score $y_s^t$ exceeding the pre-defined threshold $\tau$, it is accepted as correct and no further candidates are sampled for step $s$. The index of the first such trial, $t^*$, is defined as:

$$t^* = \min\left\{ t \mid y_s^t > \tau, \ t \leq t_{\max} \right\}. \tag{13}$$

When no candidate exceeds the threshold within $t_{\max}$ trials, we fall back to selecting the candidate with the highest score, whose index $t'$ is given by

$$t' = \arg\max_{1 \leq t \leq t_{\max}} y_s^t. \tag{14}$$

Overall, the acceptance of step $s$ can be formalized as $\hat{r}_s = r_s^{\hat{t}}$, where $\hat{t}$ denotes the accepted index:

$$\hat{t} = \begin{cases} t^*, & \text{if } \left\{ t \mid y_s^t > \tau, \ t \leq t_{\max} \right\} \neq \varnothing, \\ t', & \text{otherwise.} \end{cases} \tag{15}$$

By iteratively applying this adaptive selection strategy across all reasoning steps, the system effectively constructs a complete and adapted reasoning path $\hat{\boldsymbol{r}} = (\hat{r}_1, \ldots, \hat{r}_l)$ for a given question $q$. The resulting path is inherently more reliable, as this process evaluates and selects each step to proactively avoid erroneous reasoning in the face of domain shifts, thereby enhancing the final output's robustness. The algorithmic description of LADA is presented in Algorithm 1.

---

**Algorithm 1** LADA Algorithm

---

**Require:** Black-box LLM $\pi_{\text{LLM}}$, pretrained adapter $f_{\boldsymbol{\theta}}$, publicly available datasets $\{\mathcal{D}_i\}_{i=1}^a$, meta batch size $z$, paired few-shot example set $\mathcal{B}_T$;
    `# Task construction and meta training:`
  1: Construct meta-training task set $\mathcal{M}$ from $\{\mathcal{D}_i\}_{i=1}^a$ via clustering and error synthesis;
  2: **while** not converged **do**
  3:    Sample $z$ tasks $\{\mathcal{T}_i\}_{i=1}^z$ from $\mathcal{M}$;
  4:    **for all** $\mathcal{T}_i$ **do**
  5:        Sample a support set $\mathcal{S}_i$ and a query set $\mathcal{Q}_i$ from task $\mathcal{T}_i$;
  6:        Compute adapted parameters $\boldsymbol{\theta}_i'$ using Eq. (5) based on $\mathcal{S}_i$;
  7:    **end for**
  8:    Update $\boldsymbol{\theta}$ using Eq. (7) based on $\{\mathcal{Q}_i\}_{i=1}^z$;
  9: **end while**
    `# Test-time adaptation:`
10: Obtain the adapted parameters $\hat{\boldsymbol{\theta}}$ using Eq. (10) based on $\mathcal{B}_T$;
11: **for** $q \sim p_T(q)$ **do**
12:    Obtain adapted response $\hat{\boldsymbol{r}}$ step by step using Eq. (11) and Eq. (15);
13: **end for**
**Ensure:** Adapted responses $\hat{\boldsymbol{r}}$ for $q \sim p_T(q)$.

---

### 3.3 THEORETICAL ANALYSIS

We model the stepwise reasoning process as a Markov decision process (Puterman, 1994; Sutton & Barto, 2018; Wang, 2025). Each state $S$ is a partially constructed reasoning sequence, with $\mathbb{S}$ denoting the state space. An action $A \in \mathbb{A}$ is defined as the policy $\pi$'s generation and selection of the next reasoning step given $S$, which is appended to the current reasoning sequence to form the next state. The reward function $R(S, A)$ measures the immediate reward of an action $A$ given the current state $S$. In our method, it is instantiated by the output score of the adapter, which is trained to align with the final task objective and therefore provides a surrogate signal for the true reward.

We denote by $\pi_0$ the baseline policy that naively accepts the first sampled reasoning step at test time, and by $\pi_\tau$ our adaptive selection TTA policy. The value function of a policy $\pi$ is defined as $V_\pi(S) = \mathbb{E}_{A \sim \pi(\cdot|S)} [R(S, A) + \gamma V_\pi((S, A))]$, where $\gamma \in (0, 1)$ is a discount factor. This represents the expected cumulative reward starting from state $S$ and captures the overall reasoning quality of policy $\pi$. The associated Bellman operator is $T_\pi V(S) = \mathbb{E}_{A \sim \pi(\cdot|S)} [R(S, A) + \gamma V((S, A))]$, and the Q-function is defined as $Q_\pi(S, A) = R(S, A) + \gamma V_\pi((S, A))$, which evaluates the expected reward of taking action $A$ in state $S$ and subsequently following $\pi$. To proceed with our analysis, we introduce the following assumption:

**Assumption 1** *For any state $S \in \mathbb{S}$ and candidate actions $A_1, A_2 \sim \pi_0(\cdot \mid S)$, if $R(S, A_1) \geq R(S, A_2)$, then*

$$V_{\pi_0}((S, A_1)) \geq V_{\pi_0}((S, A_2)). \tag{16}$$

Assumption 1 means that taking an action with a higher reward leads to more favorable downstream trajectories under the baseline policy. This is reasonable when the reward reflects progress toward the final task, and actions with lower rewards are more likely to take the model in the wrong reasoning direction. With this condition in place, we now present the following theorem.

**Theorem 1** *Under Assumption 1, the adaptive resampling policy $\pi_\tau$ guarantees an expected value that is equal to or higher than that of the baseline policy $\pi_0$ at every state:*

$$V_{\pi_\tau}(S) \geq V_{\pi_0}(S) \quad \forall S \in \mathbb{S}. \tag{17}$$

The proof of Theorem 1 is provided in Appendix A. Theorem 1 guarantees the safety of the adaptive resampling policy $\pi_\tau$, ensuring reasoning quality is preserved or improved relative to the baseline policy $\pi_0$, and thus provides a rigorous theoretical foundation for its use in TTA of black-box LLMs.

Table 1: Reasoning accuracy of comparing methods on three reasoning datasets.

| Methods | Model | GSM8K | StrategyQA | ScienceQA |
|---|---|---|---|---|
| ZERO-SHOT COT | | $81.58 \pm 0.69$ | $64.92 \pm 2.09$ | $72.20 \pm 0.71$ |
| COT PROMPTING | | $83.19 \pm 0.27$ | $66.08 \pm 1.79$ | $76.00 \pm 0.99$ |
| SELF-CONSISTENCY | Qwen2-7B | $83.76 \pm 0.29$ | $66.96 \pm 0.82$ | $76.07 \pm 0.66$ |
| BBOX-ADAPTER | | $78.45 \pm 0.31$ | $68.56 \pm 0.94$ | $76.67 \pm 0.09$ |
| LADA | | $\mathbf{84.10 \pm 0.38}$ | $\mathbf{69.73 \pm 0.67}$ | $\mathbf{80.20 \pm 0.35}$ |
| ZERO-SHOT COT | | $69.04 \pm 0.50$ | $55.31 \pm 1.10$ | $71.07 \pm 0.25$ |
| COT PROMPTING | | $68.49 \pm 1.33$ | $57.35 \pm 2.29$ | $78.47 \pm 0.93$ |
| SELF-CONSISTENCY | Mixtral-8x7B | $69.22 \pm 0.30$ | $58.23 \pm 0.67$ | $78.60 \pm 0.35$ |
| BBOX-ADAPTER | | $68.03 \pm 0.49$ | $60.26 \pm 1.31$ | $76.73 \pm 0.83$ |
| LADA | | $\mathbf{70.05 \pm 0.83}$ | $\mathbf{63.51 \pm 0.74}$ | $\mathbf{80.07 \pm 0.25}$ |

## 4 EXPERIMENTS

### 4.1 EXPERIMENTAL CONFIGURATIONS

**Datasets.** We evaluate LADA on three question-answering benchmarks. GSM8K (Cobbe et al., 2021) is a math reasoning dataset where solving each problem requires multi-step reasoning. StrategyQA (Geva et al., 2021) is an implicit reasoning dataset that challenges models to infer unstated assumptions. ScienceQA (Lu et al., 2022) is a science-domain reasoning benchmark, organized into three categories: natural science, social science, and language science. Complete dataset details are given in Appendix B.1.

**Baselines.** We compare our method with four black-box approaches, including:

- ZERO-SHOT COT (Kojima et al., 2022): A prompt-based approach that instructs the model to "think step by step" at test time to derive the final answer.
- COT PROMPTING (Wei et al., 2022): A prompt-based approach that augments the prompt with chain-of-thought examples to guide multi-step reasoning.
- SELF-CONSISTENCY (Wang et al., 2023b): A decoding-based approach that samples multiple reasoning paths and aggregates them by majority voting to obtain the final answer.
- BBOX-ADAPTER (Sun et al., 2024b): A training-based approach that learns a scoring model on the target-domain training set and leverages it to guide beam search for the final prediction.

**Settings.** We consider two experimental settings: fixed target-domain TTA and changing target-domain TTA. In the first setting, we select one dataset from GSM8K, StrategyQA, or ScienceQA as the target domain, while the remaining two serve as available datasets. In the second setting, we regard the three subsets of ScienceQA (natural science, social science, and language science) as dynamically changing target domains, and use GSM8K and StrategyQA to meta-train the adapter.

**Implementation Details.** For the black-box LLM, we simulate API-style behavior using two representative open-source models: Qwen2-7B-Instruct (Yang et al., 2024), a dense decoder-only model, and Mixtral-8x7B-Instruct (Jiang et al., 2024), a sparse Mixture-of-Experts model. As the adapter, we employ DeBERTa-v3-large (He et al., 2020), which contains 0.3B parameters.

For task construction, we follow the taxonomy of reasoning errors proposed by Golovneva et al. (2023), from which we adopt $h = 7$ error types. For each available dataset, we partition the data into $k = 10$ semantic clusters using embeddings obtained from a pretrained LLM encoder. Consequently, for each target domain, we derive $u = 140$ meta-training tasks. Further details on task construction are provided in Appendix B.2.

For meta-training, we run for 10 epochs. In each epoch, we iterate over all tasks with meta-batch size $b = 8$. For each task, we use a support set of size $c = 3$ and a query set of size $d = 10$. The inner update uses 3 gradient steps with step size $\alpha = 5 \times 10^{-6}$ and is performed with SGD, while the outer update uses 1 gradient step with step size $\beta = 5 \times 10^{-6}$, optimized by AdamW (Loshchilov & Hutter, 2019) with a weight decay of 0.01. We set the margin $\zeta = 0.5$.

Table 2: Reasoning accuracy of comparing methods under domain changes of ScienceQA.

| Methods | Model | Natural | Social | Language |
|---|---|---|---|---|
| ZERO-SHOT COT | | $71.21 \pm 1.06$ | $83.84 \pm 1.43$ | $74.86 \pm 2.47$ |
| COT PROMPTING | | $74.61 \pm 1.12$ | $90.91 \pm 4.95$ | $77.44 \pm 1.08$ |
| SELF-CONSISTENCY | Qwen2-7B | $74.89 \pm 0.85$ | $91.92 \pm 1.75$ | $77.72 \pm 0.65$ |
| BBOX-ADAPTER | | $75.31 \pm 1.16$ | $90.91 \pm 3.03$ | $78.16 \pm 1.74$ |
| LADA | | $\mathbf{78.44 \pm 0.87}$ | $\mathbf{92.93 \pm 1.43}$ | $\mathbf{81.47 \pm 1.41}$ |
| ZERO-SHOT COT | | $65.39 \pm 0.20$ | $85.86 \pm 3.78$ | $74.43 \pm 1.33$ |
| COT PROMPTING | | $77.02 \pm 2.41$ | $91.92 \pm 1.75$ | $77.87 \pm 2.69$ |
| SELF-CONSISTENCY | Mixtral-8x7B | $77.44 \pm 0.85$ | $92.93 \pm 1.74$ | $78.01 \pm 0.75$ |
| BBOX-ADAPTER | | $76.31 \pm 1.91$ | $91.92 \pm 4.62$ | $77.16 \pm 1.72$ |
| LADA | | $\mathbf{79.57 \pm 1.93}$ | $\mathbf{94.95 \pm 1.43}$ | $\mathbf{82.33 \pm 0.93}$ |

Table 3: Reasoning accuracy of LADA and its variants on `ScienceQA` dataset.

| Methods | Acc. (%) |
|---|---|
| LADA-NS | $75.93 \pm 0.78$ |
| LADA-NM | $76.47 \pm 0.84$ |
| LADA-NA | $77.73 \pm 0.62$ |
| LADA | $\mathbf{80.20 \pm 0.35}$ |

Table 4: Adaptation time and 10-sample inference time, evaluated for LADA and baselines on `StrategyQA` dataset.

| Methods | Adapt. (s) | Infer. (s) |
|---|---|---|
| COT PROMPTING | - | 8.49 |
| SELF-CONSISTENCY | - | 86.35 |
| BBOX-ADAPTER | 4.15 | 46.49 |
| LADA | 2.61 | 20.54 |

For test-time adaptation, we perform 3 update steps using AdamW, with the adaptation step size $\gamma = 1 \times 10^{-6}$. Only the last four layers and the classification head of the adapter are updated. We set the acceptance threshold $\tau = 0.5$ and the maximum number of sampling attempts $t_{\max} = 5$. For the baselines, the number of reasoning paths sampled for SELF-CONSISTENCY is set to 10, and BBOX-ADAPTER is run with its default parameters. Each baseline is given access to the same 3 target-domain examples, except for ZERO-SHOT COT.

## 4.2 EXPERIMENTAL RESULTS

We conduct 3 trials with different random seeds, reporting both the mean and standard deviation of the reasoning accuracy, and the results are summarized in Table 1 and Table 2. The best performance is shown in boldface, and the second-best result is underlined. The results show that:

- LADA consistently achieves the best performance across three benchmark datasets in two settings, outperforming all baseline approaches.
- In the fixed target-domain TTA setting, LADA performs meta-training with different combinations of available datasets, yielding average target-domain improvements of $3.89\%$ on Qwen2-7B-Instruct and $4.61\%$ on Mixtral-8x7B-Instruct over COT PROMPTING, showing its effectiveness in transferring knowledge from varied sources to new targets.
- In the changing target-domain TTA setting, LADA leverage the meta-trained adapter and improve performance on dynamic target domains, with an average performance gain of $4.06\%$ on Qwen2-7B-Instruct and $4.07\%$ on Mixtral-8x7B-Instruct compared with COT PROMPTING, demonstrating that the meta-trained adapter generalizes effectively across dynamic target domains.

## 4.3 FURTHER ANALYSIS

To verify the effectiveness of the components in LADA, we conduct an ablation study with three vanilla variants: LADA-NS, LADA-NM, and LADA-NA. In LADA-NS, the first sampled reasoning step is directly accepted without selection. In LADA-NM, the meta-training procedure is removed, and the adapter is trained with supervised learning. In LADA-NA, we directly apply the adapter trained with meta-learning, without further adapting it to the target domain. We evaluate these

Table 5: GPU memory consumption during adaptation phase and reasoning accuracy of LADA and compared methods on `StrategyQA` dataset.

| Methods | Mem. (GiB) | Acc. (%) |
|---------|-----------|----------|
| LADA | 3.87 | 80.20 |
| LADA-ALL | 7.83 | 79.80 |
| LLM-FT | 15.14 | 76.07 |

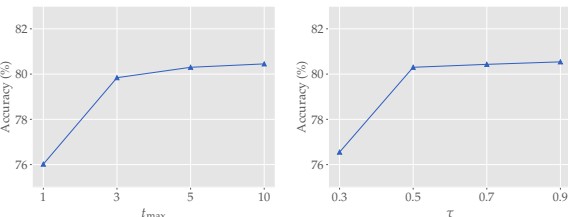

Figure 1: Sensitivity analysis of $t_{max}$ and $\tau$.

variants on `StrategyQA` using Qwen2-7B-Instruct, and results are reported in Table 3. The results show that LADA-NS and LADA-NM suffer substantial performance degradation, highlighting the importance of both selection and meta-training. For LADA-NA, although meta-training enables the adapter to acquire transferable adaptation skill, its performance still falls short of LADA, which further adapts to the target domain.

To evaluate the efficiency of LADA, we measured the adaptation and inference time of LADA and the baselines on `StrategyQA` using Qwen2-7B-Instruct. Inference time was evaluated on a subset of 10 samples to simulate a single conversation with thematically related questions, as is typically the case in real-world scenarios (Deng et al., 2023). We invoke `torch.cuda.synchronize` before measurement to guarantee that reported computational overhead reflects completed GPU operations. The results are presented in Table 4. It can be observed that, compared with inference time, the adaptation time of LADA accounts for only a small fraction. Moreover, thanks to its adaptive selection mechanism, LADA requires less than one-quarter of the time used by SELF-CONSISTENCY for full-answer resampling, highlighting its efficiency.

We further evaluate the GPU memory consumption of our approach. To this end, we introduce a variant, LADA-ALL, which updates all adapter parameters during adaptation. In addition, we include a baseline, LLM-FT, which directly accesses the LLM parameters and fine-tunes them with LoRA (Hu et al., 2022) on the same 3 examples for consistency. We measure peak GPU memory usage with `torch.cuda.max_memory_allocated` during the adaptation phase. The GPU memory consumption and reasoning accuracy of the three methods, evaluated on `ScienceQA` with Qwen2-7B-Instruct, are reported in Table 5. The results show that LADA consumes the least GPU memory, whereas LLM-FT requires nearly four times more memory than LADA, making it impractical in resource-constrained scenarios. Moreover, the few-shot setting hinders LLM-FT from achieving strong generalization. Interestingly, LADA also outperforms LADA-ALL, suggesting that full adapter updates in the few-shot setting may lead to overfitting, whereas restricting updates to the top layers helps preserve the generalization learned during meta-training.

Lastly, we study the sensitivity of two test-phase hyperparameters in LADA, $t_{max}$ and $\tau$, on `ScienceQA` with Qwen2-7B-Instruct. The results are presented in Fig.1. We observe that when $t_{max} \geq 3$, the performance continues to improve with larger values of $t_{max}$ but remains overall stable, indicating that LADA can correct faulty reasoning steps with only a small number of resampling attempts. Since we formulate the decision of whether the next reasoning step is acceptable as a binary classification problem during training, $\tau = 0.5$ is a natural choice. Setting $\tau$ below this threshold tends to introduce erroneous reasoning steps, while higher thresholds yield marginal performance gains at the cost of repeatedly resampling correct steps, thus reducing overall efficiency.

## 5 CONCLUSION

In this paper, we propose a novel framework LADA for continuous, rapid adaptation of black-box LLMs to unseen, dynamic domains at test time. LADA meta-trains an adapter on diverse tasks from multiple datasets (covering semantic clusters and error types) to learn transferable adaptation skills. At test time, the adapter uses a small set of current-domain positive–negative reasoning pairs for lightweight adaptation, then guides the frozen LLM step-by-step to select reliable reasoning steps, steering generation toward domain-appropriate trajectories without parameter updates or prior domain exposure. Experiments on various benchmark datasets under two settings validate the effectiveness of the proposed approach.

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

# A  PROOFS

## A.1  USEFUL LEMMAS

**Lemma 1** *For any policy $\pi$, the Bellman operator $T_\pi$ is a $\gamma$-contraction on $(\mathbb{R}^{\mathbb{S}}, \|\cdot\|_\infty)$. Therefore, there exists a unique fixed point $V_\pi$ such that $T_\pi V_\pi = V_\pi$.*

*Proof.* Let $V_1, V_2 \in \mathbb{R}^{\mathbb{S}}$. For any fixed state $S$,

$$
\begin{aligned}
|T_\pi V_1(S) - T_\pi V_2(S)| &= \big| \mathbb{E}_{A \sim \pi(\cdot|S)} \left[ R(S, A) + \gamma V_1((S, A)) \right] \\
&\quad - \mathbb{E}_{A \sim \pi(\cdot|S)} \left[ R(S, A) + \gamma V_2((S, A)) \right] \big| \\
&= \big| \mathbb{E}_{A \sim \pi(\cdot|S)} \left[ \gamma(V_1((S, A)) - V_2((S, A))) \right] \big| \\
&\leq \gamma \cdot \mathbb{E}_{A \sim \pi(\cdot|S)} \left[ |V_1((S, A)) - V_2((S, A))| \right] \\
&\leq \gamma \cdot \|V_1 - V_2\|_\infty.
\end{aligned}
\tag{18}
$$

Taking supremum over all $S$ yields:

$$
\|T_\pi V_1 - T_\pi V_2\|_\infty \leq \gamma \|V_1 - V_2\|_\infty,
\tag{19}
$$

showing that $T_\pi$ is a $\gamma$-contraction mapping with $\gamma < 1$.

Since $(\mathbb{R}^{\mathbb{S}}, \|\cdot\|_\infty)$ is a complete metric space, Banach's fixed-point theorem (Puterman, 1994) ensures that $T_\pi$ admits a unique fixed point $V_\pi \in \mathbb{R}^{\mathbb{S}}$ satisfying $T_\pi V_\pi = V_\pi$, and the iteration $V_{k+1} = T_\pi V_k$ converges to $V_\pi$ for any initialization $V_0$. $\qquad\square$

**Lemma 2** *Given two policies $\pi$ and $\pi'$, if $T_{\pi'} V_\pi(S) \geq V_\pi(S) \ \forall S \in \mathbb{S}$, then it follows that $V_{\pi'}(S) \geq V_\pi(S) \ \forall S \in \mathbb{S}$.*

*Proof.* From the Bellman equation, we obtain

$$
\begin{aligned}
T_{\pi'} V_{\pi'}(S) &= \mathbb{E}_{A \sim \pi'(\cdot|S)} \left[ R(S, A) + \gamma V_{\pi'}((S, A)) \right] \\
&= \mathbb{E}_{A \sim \pi'(\cdot|S)} \left[ R(S, A) \right] + \gamma \mathbb{E}_{S' \sim P_{\pi'}(\cdot|S)} V_{\pi'}(S') \\
&= \mathbb{E}_{A \sim \pi'(\cdot|S)} \left[ R(S, A) \right] + \gamma P_{\pi'} V_{\pi'}(S),
\end{aligned}
\tag{20}
$$

where $P_{\pi'}$ is the transition kernel, representing the conditional distribution over the next state $S'$ given the current state $S$ when policy $\pi'$ is applied.

From Lemma 1, we have

$$
V_{\pi'}(S) = T_{\pi'} V_{\pi'}(S) = R_{\pi'}(S) + \gamma P_{\pi'} V_{\pi'}(S),
\tag{21}
$$

where we write $R_{\pi'}(S) = \mathbb{E}_{A \sim \pi'(\cdot|S)}[R(S, A)]$ as the expected one-step reward under policy $\pi'$ for brevity. Define $\Delta(S) = V_{\pi'}(S) - V_\pi(S)$. Then we obtain

$$
\begin{aligned}
\Delta &= V_{\pi'} - V_\pi \\
&= R + \gamma P_{\pi'} V_{\pi'} - V_\pi \\
&= R + \gamma P_{\pi'} V_\pi + \gamma P_{\pi'}(V_{\pi'} - V_\pi) - V_\pi \\
&= T_{\pi'} V_\pi + \gamma P_{\pi'} \Delta - V_\pi,
\end{aligned}
\tag{22}
$$

which rearranges to

$$
(I - \gamma P_{\pi'})\Delta = T_{\pi'} V_\pi - V_\pi.
\tag{23}
$$

By the premise of this lemma, the right-hand side is non-negative.

Since $P_{\pi'}$ is a stochastic kernel, it defines a positive contraction on $(\mathbb{R}^{\mathbb{S}}, \|\cdot\|_\infty)$, hence $\|\gamma P_{\pi'}\|_\infty \leq \gamma < 1$. Therefore $I - \gamma P_{\pi'}$ is invertible with the Neumann series $(I - \gamma P_{\pi'})^{-1} = \sum_{k \geq 0} (\gamma P_{\pi'})^k$, which is a positive operator (Horn & Johnson, 2012). Since $T_{\pi'} V_\pi(S) - V_\pi(S) \geq 0$ for all $S$, and $(I - \gamma P_{\pi'})^{-1}$ is a positive operator, applying it preserves non-negativity, hence $\Delta(S) \geq 0$ for all $S$. Therefore $V_{\pi'}(S) \geq V_\pi(S)$ for all $S$. $\qquad\square$

## A.2 Proof of Theorem 1

*Proof.* Let $A_0 \sim \pi_0(\cdot|S)$ be the baseline action, and $A_\tau \sim \pi_\tau(\cdot|S)$ be the action of adaptive resampling policy. We have

$$
\begin{aligned}
&\mathbb{E}[Q_{\pi_0}(S, A_\tau)] - \mathbb{E}[Q_{\pi_0}(S, A_0)] \\
&= \mathbb{E}[R(S, A_\tau) - R(S, A_0)] + \gamma \, \mathbb{E}[V_{\pi_0}((S, A_\tau)) - V_{\pi_0}((S, A_0))] \\
&\geq 0.
\end{aligned}
\tag{24}
$$

The reward term is nonnegative since the adaptive resampling policy never accepts a lower-reward action than the baseline, and the value term is nonnegative by Assumption 1. So,

$$
\mathbb{E}[Q_{\pi_0}(S, A_\tau)] \geq \mathbb{E}[Q_{\pi_0}(S, A_0)] = V_{\pi_0}(S).
\tag{25}
$$

Using the Bellman operator $T_{\pi_\tau} V_{\pi_0}(S) = \mathbb{E}[Q_{\pi_0}(S, A_\tau)]$, we have

$$
T_{\pi_\tau} V_{\pi_0}(S) \geq V_{\pi_0}(S) \quad \forall S \in \mathbb{S}.
\tag{26}
$$

From Lemma 2, since $T_{\pi_\tau} V_{\pi_0}(S) \geq V_{\pi_0}(S)$, we conclude:

$$
V_{\pi_\tau}(S) \geq V_{\pi_0}(S) \quad \forall S \in \mathbb{S}.
\tag{27}
$$

This finishes the proof. □

## B Experimental Details

### B.1 Additional Dataset Details

Our evaluation spans three benchmarks: `GSM8K` for mathematical reasoning, `StrategyQA` for commonsense inference, and `ScienceQA` for scientific reasoning, covering distinct domains and reasoning paradigms. Details of these datasets are summarized below:

- `GSM8K` (Cobbe et al., 2021) contains 8.5K grade school math word problems, with 7.5K for training and 1K for testing. Problems require 2–8 reasoning steps using basic arithmetic. Written by human annotators with quality control, solutions are provided in natural language, supporting interpretable step-by-step reasoning evaluation of LLMs.
- `StrategyQA` (Geva et al., 2021) contains 2,288 yes/no questions, with 2,059 for training and 229 for testing, targeting implicit multi-step reasoning. Unlike explicit multi-hop datasets, reasoning steps are not given but inferred as strategies. Each question is short, diverse, and linked to supporting Wikipedia evidence, covering a broad range of domains and reasoning types.
- `ScienceQA` (Lu et al., 2022) is a multimodal multiple-choice benchmark of 21,208 science questions across natural, social, and language sciences, with text and image contexts. Each question includes lectures and explanations for reasoning evaluation. Following Sun et al. (2024a), we excluded image-based questions and sampled 2,000 training and 500 testing questions from the original splits for our experiments.

### B.2 Additional Task Construction Details

Before synthesizing data for meta-training, we obtain step-by-step reasoning traces for the training splits of the relevant datasets. `GSM8K` directly provides annotated step-by-step reasoning traces, whereas for `StrategyQA` and `ScienceQA`, which lack such annotations, we employ DeepSeek-V3 (DeepSeek-AI, 2024) to generate corresponding traces and discard erroneous samples.

To better simulate the error types that arise under distribution shift, we employ DeepSeek-V3 to synthesize seven types of reasoning error types frequently observed in LLMs (Golovneva et al., 2023): factuality, hallucination, redundancy, repetition, missing step, coherency, and commonsense. To ensure the rationales and formats of the generated samples remain consistent, we instruct the LLM to verify its own outputs and provide justifications. The prompts used for error synthesis are as follows.

---

**Prompt used for data synthesis**

```
You are a data synthesis assistant.

Your input is a correct reasoning process, which can be formalized as
follows:  Q: question A: Let's think step by step.  r_start, ...  ,r_i,
...  ,r_end, where r denotes a reasoning step.

First, you need to **randomly select** any reasoning step r_i from the
correct reasoning steps.
Then:
- Generate a correct version right(r_i) by rewriting the original step
r_i in a different form, while fully preserving its factual meaning.
- Generate a corresponding faulty version wrong(r_i) by introducing an
error into r_i.
Note:
- The type of error you need to synthesize is:  **Coherency**, which
refers to:  **Steps contradict each other or do not follow a cohesive
story**.
- The faulty step should realistically simulate the **Coherency**
mistake that large language models are likely to make during
reasoning and should be significant enough to affect the subsequent
reasoning.

Next, you need to construct a positive and a negative sample.  You
only need to include the reasoning steps up to and including the
synthesized correct step right(r_i) (for the positive sample), or up
to and including the synthesized incorrect step wrong(r_i) (for the
negative sample).  Do not include the rest of the reasoning chain.

The positive sample can be formalized as:  " Q: question A: Let's
think step by step.  r_start, ...  ,right(r_i) # the synthesized right
reasoning step "

The negative sample can be formalized as:  " Q: question A: Let's
think step by step.  r_start, ...  ,wrong(r_i) # the synthesized wrong
reasoning step "

Finally, in terms of output format, please return the positive and
negative samples in a list.  Each sample should be a string, and each
reasoning step should occupy one line, as follows:  [ " Q: question
A: Let's think step by step.  r_start ...  right(r_i) ", " Q: question A:
Let's think step by step.  r_start ...  wrong(r_i) " ]

Before providing the list, briefly explain the rationale behind the
construction of positive and negative samples, and how the negative
(faulty) samples may affect subsequent reasoning.  At last, please
check whether your output format meets the specified requirements.

Your task:
```

## B.3 CASE STUDY

**Q:** Could you go to New York Public Library and the Six Flags Great Escape in the same day?
**A:** New York Public Library is in Manhattan, New York City. (0.92, →)
Six Flags Great Escape is located in Lake George, New York. (0.87, →)
New York City and Lake George are in different states and far apart. (0.24, ↻)
The average driving time between Manhattan and Lake George is around 5-6 hours, depending on traffic. (0.76, →)
#### Yes.

Figure 2: Case study of LADA on StrategyQA, where parentheses show (score, action); → denotes moving to the next step, and ↻ denotes regenerating the current step.

Figure 2 presents a case study on the `StrategyQA` dataset, where the original model generates an erroneous reasoning step that would mislead the solution, but LADA intervenes to revise the step and successfully guides the reasoning trajectory to the correct answer.

## C LLMs Usage

LLMs were used exclusively to assist with writing and polishing the manuscript. They helped refine language, improve readability, and enhance clarity through tasks such as sentence rephrasing, grammar checking, and improving the overall flow of the text.

The LLM was not involved in ideation, research methodology, or experimental design. All scientific concepts, analyses, and conclusions were developed solely by the authors. The authors take full responsibility for the content of the manuscript, and the use of the LLM adhered to ethical guidelines without contributing to plagiarism or scientific misconduct.

