# OpenReview forum: "LADA: Enabling Adaptation of Black-Box LLMs to Dynamic Domain Changes at Test Time"
_ICLR.cc/2026/Conference — ICLR 2026 Conference Withdrawn Submission_

### Official Review · Reviewer_H2sx · 2025-10-21

**Soundness:** 2
**Presentation:** 3
**Contribution:** 2
**Rating:** 4
**Confidence:** 3

**Summary:**

This paper introduces LADA, a meta-learning-based framework designed to enable black-box large language models (LLMs) to adapt to dynamic domain shifts at test time. The method meta-trains an adapter across diverse datasets and error types so that, during inference, the adapter can be quickly adapted using a few target-domain examples to guide the LLM reasoning process step by step.

**Strengths:**

1.	Addresses the practical problem of adapting black-box LLMs without parameter access.
2.	Presents a clear and well-organized description of the proposed meta-learning framework.
3.	Includes theoretical analysis ensuring that the adaptive selection policy does not degrade performance compared to a baseline.

**Weaknesses:**

1.	LADA test-time procedure assumes access to paired positive–negative target-domain examples for “lightweight adaptation” and evaluates on predefined full datasets/subsets rather than streaming unlabeled batches, which deviates from the canonical unsupervised, online TTA setting.
2.	Although the paper emphasizes efficiency at test time, the meta-training phase involves extensive task construction (e.g., clustering, error synthesis) and iterative inner-outer loop optimization. This process is computationally intensive and requires access to multiple datasets and a powerful oracle LLM, which may limit the practicality of LADA for resource-constrained researchers or applications.
3.	The experiments are conducted only on three relatively small reasoning datasets (GSM8K, StrategyQA, and ScienceQA). To convincingly demonstrate robustness under diverse domain shifts, the authors should evaluate LADA on more challenging and large-scale benchmarks such as AIME24, MATH-500, and AdaptEval [A], which are explicitly designed to test reasoning and adaptation capabilities across domains.
4.	Since the paper focuses on black-box LLM test-time adaptation, it is essential to include experiments on closed-source API-based LLMs (e.g., GPT-3.5-turbo, Claude), as done in BBOX-ADAPTER [B]. Without such experiments, the practicality of LADA for real-world black-box adaptation remains unverified.
5.	Why the selection of DeBERTa-v3-large as the adapter model?

[A] Test-Time Learning for Large Language Models, ICML 2025.

[B]  Bbox-adapter: Lightweight adapting for black-box large language models, ICML 2024.

**Questions:**

see Weaknesses.

---

### Official Review · Reviewer_zLAd · 2025-10-28

**Soundness:** 3
**Presentation:** 1
**Contribution:** 2
**Rating:** 2
**Confidence:** 3

**Summary:**

This paper introduces LADA, a meta-learning framework for test-time adaptation of black-box large language models (LLMs) under dynamic domain shifts. LADA trains an external adapter on a diverse set of reasoning tasks and error types, enabling few-shot adaptation at test time without modifying LLM parameters (but update the scoring function). The adapter scores step-level reasoning candidates and adaptively accepts or resamples them to improve domain robustness. The authors provide a simple theoretical guarantee of non-decreasing cumulative reward and demonstrate consistent gains across reasoning datasets (GSM8K, StrategyQA, ScienceQA) with Qwen2-7B and Mixtral-8×7B models.

**Strengths:**

1. Addresses a timely and practical problem which is the black-box test-time adaptation, in the context of dynamic environment.

2. The integration of meta-learning with step-level reasoning control is intuitive and well-motivated.

3. LADA achieves consistent improvements over strong baselines across multiple datasets and models.

4. Good amount of ablations and good amount of experiments.

**Weaknesses:**

1. The adapter’s meta-training relies on an oracle LLM to synthesize positive/negative reasoning pairs, not sure how well this transform in reality.

2. All benchmarks are reasoning QA datasets; no experiments demonstrate domain adaptation beyond QA.

3. The theoretical contribution is very limited, not sure how much section 3.3. help with the paper.

4. The paper is not well written without enough intuitive explanations (too much - reads like chatgpt-ish text), an example of how adaptation change before and after in terms of reasoning trace will be very helpful.

5. The “changing domain” setup (ScienceQA subsets) is a weak proxy for changing environments, not sure how much I am convinced with this setup. Maybe some visualisation of the reasoning trace under different subsection help.

**Questions:**

In general, I think the idea is not bad but the paper seems to be in a very early stage and need some major revision and more experimental results. Please refer to the weakness section.

---

### Official Review · Reviewer_KdFQ · 2025-10-28

**Soundness:** 3
**Presentation:** 2
**Contribution:** 3
**Rating:** 4
**Confidence:** 3

**Summary:**

The paper proposes a novel framework named LADA for continuous, rapid test-time adaptation of black-box large language models (LLMs) to unseen and dynamically changing domains. LADA enables adaptation without accessing or updating the LLM parameters, a key challenge for API-served black-box models. It leverages meta-trained adapters that learn transferable adaptation skills across diverse datasets and error types. At test time, the adapter uses a few target-domain examples for lightweight task-specific adaptation and guides the LLM stepwise through adaptive selection of reliable reasoning steps, steering generation toward domain-appropriate trajectories.

**Strengths:**

1. The problem studied is important and novel.

2. Leveraging meta-learning on diverse tasks teaches transferable adaptation skills, addressing the data scarcity concerns in target domain.

3. Overall LADA provides a robust solution for test time adaptation to various domains.

**Weaknesses:**

1. The meta training relies on the synthesized reasoning error types and clustering which might not reflect the real world scenarios very well.

2. The method is computationally intensive where multiple trials for each step are made which are then verified, this makes the method computationally very heavy.

3. The setup of oracle LLM is not clear, what does it look like is not specified in the paper.

4. While small in size as compared to LLM, the adapter still has considerable computational overhead with 0.3 B params.

5. In Eq. 2 why a sum is required when the indicator will fire only at the time when the threshold is met and as soon as the threshold is met the sampling stops. It makes the setup bit confusing.

**Questions:**

See Weaknesses.

---

### Official Review · Reviewer_YSeg · 2025-10-29

**Soundness:** 2
**Presentation:** 2
**Contribution:** 2
**Rating:** 4
**Confidence:** 2

**Summary:**

The paper proposes LADA, a meta-learning framework to enable test-time adaptation for black-box LLMs in dynamically changing domains. LADA pre-trains an adapter on a diverse set of tasks to learn general adaptation skills, which is then rapidly fine-tuned at test time with a few examples. This adapter guides the frozen LLM's generation step-by-step, selecting reliable reasoning steps to improve performance on unseen target domains.

**Strengths:**

The paper tackles a well-defined and highly relevant problem: adapting pretrained LLMs to new domains without full retraining, a critical challenge for real-world deployment. The proposed solution of using meta-learning to learn a transferable adaptation skill is well-suited for the problem of low-data domains. The methodology is clearly explained, and the experimental setup appears thorough, with multiple benchmarks, strong baselines, and insightful ablation studies. The results presented are strong and consistently demonstrate the effectiveness of the proposed method.

**Weaknesses:**

- Reliance on oracle LLM for task construction: The meta-training task construction critically relies on an oracle LLM to synthesize positive-negative reasoning pairs and reasoning errors types. This dependency raises concerns about the quality, completeness, and generalizability of the meta-training data; if the oracle LLM fails to synthesize realistic or complex errors, the adapter's learned transferability might be artificially limited.
- While LADA aims for adaptation to unseen domains, the meta-training data is drawn from a fixed set of publicly available datasets. The effectiveness of transferring skills to domains truly outside the scope and error types simulated by these source datasets remains unverified, especially in complex, real-world domain shifts.
- Limited ablation on meta-learning components: The ablation study only broadly validates the meta-training process. A more fine-grained ablation on the inner-loop loss and the effectiveness of clustering and specific error types would provide deeper insight into which meta-training elements are most critical for transferability.

**Questions:**

- How realistic is Assumption 1? In complex multi-step reasoning, an immediately "high-reward" (high-scoring) step might still lead to a dead end later if the oracle/adapter is locally shortsighted, raising questions about whether the assumption holds universally in practice.
- Oracle robustness: How sensitive is the final performance of LADA to the choice and quality of the oracle LLM?
- Domain boundaries: The paper evaluates dynamic shifts among subsets of ScienceQA. Could the authors test LADA on a sequence of TTA tasks where the target domains are fundamentally different from the meta-training sources?

---

### Note · Authors · 2025-11-13

I have read and agree with the venue's withdrawal policy on behalf of myself and my co-authors.